# Prenatal Mercury Exposure and Infant Weight Trajectories in a UK Observational Birth Cohort

**DOI:** 10.3390/toxics11010010

**Published:** 2022-12-22

**Authors:** Kyle Dack, Robyn E. Wootton, Caroline M. Taylor, Sarah J. Lewis

**Affiliations:** 1Medical Research Council Integrative Epidemiology Unit, University of Bristol, Bristol BS8 2BN, UK; 2Nic Waals Institute, Lovisenberg Diaconal Hospital, 0771 Oslo, Norway; 3Population Health Sciences, Bristol Medical School, University of Bristol, Bristol BS8 2BN, UK; 4Centre for Academic Child Health, Bristol Medical School, University of Bristol, Bristol BS8 1NU, UK

**Keywords:** pregnancy, mercury, postnatal growth, birth cohort, ALSPAC

## Abstract

Mercury is highly toxic metal found in trace quantities in common foods. There is concern that exposure during pregnancy could impair infant development. Epidemiological evidence is mixed, but few studies have examined postnatal growth. Differences in nutrition, exposures, and the living environment after birth may make it easier to detect a negative impact from mercury toxicity on infant growth. This study includes 544 mother–child pairs from the Avon Longitudinal Study of Parents and Children. Blood mercury was measured in early pregnancy and infant weight at 10 intervals between 4 and 61 months. Mixed-effect models were used to estimate the change in infant weight associated with prenatal mercury exposure. The estimated difference in monthly weight gain was −0.02 kg per 1 standard deviation increase in Hg (95% confidence intervals: −0.10 to 0.06 kg). When restricted to the 10th decile of Hg, the association with weight at each age level was consistently negative but with wide confidence intervals. The lack of evidence for an association may indicate that at Hg levels in this cohort (median 1.9 µg/L) there is minimal biological impact, and the effect is too small to be either clinically relevant or detectable.

## 1. Introduction

Infant growth is a common concern for both parents and health professionals because it can be used to approximate overall postnatal health [1,2,3]. Children who grow at a slower rate than expected according to growth charts standardised by age and sex [4] are described as having faltering growth (or previously, failure to thrive) [5,6]. Observational evidence indicates that the consequences of slow growth early in life can include persistently lower weight and height throughout childhood [7,8], lower psychomotor, cognitive and IQ scores [7,9,10], and weakened immune systems [8]. Nutritional intake is the primary factor affecting growth during childhood [11], but growth can also be disrupted by disease, behavioural factors, and environmental exposures [12]. Toxic metals have been suggested to interfere with metabolic processes essential for physical development [13], and this may include the element mercury.

Mercury (Hg) is a metal released into the environment primarily through industrial activity [14,15] and to a lesser extent from volcanoes, geothermal vents, and other natural sources [16,17]. This may contaminate nearby water, soil, and food supplies of local populations [18]. Emissions also disperse globally through the atmosphere and into oceans where methylation into methylmercury (MeHg) can occur, which can bioaccumulate in predatory and/or long-lived fish (e.g., swordfish, bluefin tuna) [19,20]. Seafood consumption in a UK cohort was estimated to account for 9% of the variance in blood mercury, and another 11% was due to other dietary sources [21]. The relative importance of exposure sources remains uncertain, and it is likely that there are genetic factors that mediate metabolism [22].

Mercury is highly reactive with multiple toxic mechanisms that could impact infant development and growth. Methylmercury has a high affinity for selenohydrl and sulfhydrl groups that deplete antioxidant enzymes such as glutathione peroxidase, catalase, and copper/zinc superoxide dismutase [14,23,24], promoting oxidative stress, which may disrupt the cellular processes involved in child growth [25]. Other mechanisms that may affect growth include the depletion of metallothionein proteins required for copper and zinc homeostasis [26], DNA methylation [27] and cell death [28], and the inhibition of cell proliferation [29] and DNA repair [30]. Prenatal exposure is of particular relevance to infant growth because mercury can readily cross the placenta and accumulate in the infant’s body, where it may remain for a prolonged time after delivery [31]. This could impair both in utero and postnatal growth through the direct toxic effects described earlier. Mercury is also known to trigger epigenetic alterations [27], and epigenetic foetal programming is known to be a factor in both pre- and postnatal growth [32].

The risks to childhood development from mercury exposure are most clearly seen in two well documented incidents of local contamination in Iraq and Japan, where infant mortality and development were severely worsened [33,34]. However, this does not reflect the context of exposure for most individuals, namely chronic exposure to relatively low concentrations of mercury. Here the evidence of developmental harm is less clear. Epidemiological studies of mercury and growth have mostly focused on the impact on birth outcomes [35,36,37]. There are some indications that mercury is associated with a reduction in birth weight at the highest levels of chronic exposure [37,38,39,40], but this finding is not replicated in all high-exposure populations [41,42].

It is possible that prenatal mercury exposure may also impact growth in the postnatal period due to the slow clearance rate of mercury. The biological half-life of mercury—the speed at which concentrations are halved—is estimated to be between 30 and 120 days for methylmercury [43] and 30 to 60 days for inorganic mercury [44]. However, in children metabolic rates may differ, and in certain tissues, such as the brain, retention is believed to be much longer [45]. The postnatal environment may differ from the prenatal in ways which are less protective of the toxic effects of mercury, and the impact on growth may be more detectable. For example, it is possible after birth for the intake of long-chain polyunsaturated fatty acids to fall [45], a nutrient protective against Hg toxicity [45].

Two previous studies have assessed prenatal mercury and infant growth. The Mothers and Children’s Environmental Health study (MOCEH) in South Korea reported a negative relationship between maternal blood mercury and childhood weight at 12 and 24 months, with the strongest evidence for blood taken at 28–42 weeks gestation and weight at 24 months of age [46]. These findings were broadly replicated in a 2021 study in the Norwegian Mother, Father and Child Cohort Study (MoBa) of infant growth between 0 and 8 years of age [47]. The top decile of prenatal mercury exposure was associated with lower weight, height, and BMI gain over the study period, with particularly strong evidence of an association with weight from 18 months of age onwards [47].

Selenium (Se) is a known antagonist of mercury [48]: dietary sources include cereals, animal organs, and fish [49]. Experimental findings indicate that selenium can bind with mercury and remove it from circulation [50,51] in addition to selenium being a key precursor to thyroid hormones associated with growth [52]. However, epidemiological evidence that selenium has a clinically notable effect on mercury toxicity is scarce and inconsistent [53,54,55,56]. It may also be that the causal relationship is reversed so that one of the primary mechanisms of mercury toxicity is due to it creating selenium insufficiency [56].

This study aims (1) to assess whether maternal blood mercury is negatively associated with change in infant weight between 4 and 61 months in a UK birth cohort using mixed effects models, (2) to explore if the association between mercury and growth is mediated through an interaction effect with blood selenium, and (3) to determine the association between prenatal mercury and static weight at 10 time intervals from 4 to 61 months.

## 2. Materials and Methods

### 2.1. Study Population

The Avon Longitudinal Study of Parents and Children (ALSPAC) is a UK-based birth cohort designed to examine how environmental and genetic characteristics affect long-term health. The study recruited 14,541 women living in the former Avon Health Authority area with an expected delivery date between April 1991 and December 1992, resulting in a cohort of 14,062 live births [57]. A detailed cohort description is available [58] and the study website (http://www.bristol.ac.uk/alspac/researchers/our-data/ accessed on 16 November 2022) contains details of all the data that are available, including data sources and questionnaires, through a fully searchable data dictionary and variable search tool.

### 2.2. Exposure Assessment

Blood samples were obtained from 4484 women during a routine antenatal care visit early in pregnancy. Midwives used a vacutainer system to collect blood samples. The median time of collection was at 11 weeks’ gestation and interquartile range 9–13 weeks, with 93% of samples collected before 18 weeks. Whole blood samples were stored at 4 °C before being sent within 1–4 days to the central Bristol laboratory. Samples were transported at room temperature for up to 3 h and stored at 4 °C as whole blood until the time of analysis.

The mercury content of blood samples was measured using inductively coupled plasma dynamic reaction cell mass spectrometry (ICP-DRC-MS) at the Centers for Disease Control and Prevention (CDC), Bethesda, CDC method 3009.1. Quality control procedures were applied as previously described [21,59,60]. The number of valid measurements following quality control were 4131. One sample was below the limit of detection for mercury (0.24 μg/L) and was assigned a value of 0.7 times the lower limit of detection [61].

### 2.3. Outcome

A 10% sample of the ALSPAC cohort, known as the Children in Focus (CIF) group, were selected at random from births that occurred during the final 6 months of maternal recruitment to the study. Families included in the CIF sample were invited to attend clinics at the University of Bristol at approximately 4, 8, 12, 18, 25, 31, 37, 43, 49, and 61 months of age. At each clinic an array of measurements was taken, including the child’s weight to the nearest 0.010 kg. In total, 1432 children attended at least one clinic, with the total attendees per clinic time point ranging from 994 to 1314 children. Details of attendance rates per clinic time point and the procedures used to measure weight are described in a previous study [62].

Weight was not standardised by birth weight or gestational age because this risked introducing collider or mediation bias [63], and it was not necessary given that we used a multilevel approach where differences in starting weight were accounted for using a random intercept.

### 2.4. Covariates

Potential confounding factors that could be associated with both mercury exposure and infant growth were identified based on previous studies. These were maternal age (years), parity, pre-pregnancy BMI, alcohol consumption (units per week), smoking habits (cigarettes per day), socio-economic status approximated through highest level of education (none/CSE/vocational/O-level and A-level/degree), and frequency of oily fish, white fish, and shellfish consumption. Information was obtained from questionnaires completed by the mother at 8 and 32 weeks’ gestation and prior to delivery.

Selenium was identified as a both potential key competing exposure which could explain variance in growth, and possible modifier of mercury toxicity expression on growth. Selenium blood concentration measurements were obtained from the same blood samples as mercury, using ICP-DRC-MS and the same quality control measures described above [64].

Child’s exact age at the time of clinic attendance (in weeks) was used as the time variable for the multilevel models, centred at first clinic attendance. Place of residence and ethnicity were identified as potential confounders but not adjusted for because the sample did not vary by location and the proportion of participants not classified as “white” was low (2.4%).

### 2.5. Statistical Analysis

Statistical analyses were carried out in R version 4.1.0 [65].

Children born very prematurely (<33 weeks’ gestation) and from multiple pregnancies were excluded. Descriptive statistics were generated for the full ALSPAC sample, participants with mercury samples, and for the Children in Focus subsample to assess the representativeness of the included participants. Histograms of the distribution of mercury, graphs of postnatal growth patterns, a correlation matrix of all continuous variables, and boxplots of categorical variables were all produced using the *ggplot2* library [66]. The missingness of the data was evaluated by comparing participants with and without missing data, with no systematic pattern of missingness apparent (further details in Appendix A). Missing covariate values were estimated using multiple imputation with the *MICE* package in R, with 20 iterations of 9 imputations.

Mixed-effect modelling was used to estimate the associations between mercury and postnatal growth, using repeated assessments of weight from the same children across early childhood. Mercury was standardised to 1 standard deviation unit, which was 1.048 μg/L. A random intercept allowed for variation in starting weight amongst children. A random slope allowed for individual differences in growth trajectory across child age. Models were adjusted for pre-specified covariates likely to confound the exposure-outcome relationships of interest, based on evidence from prior studies and the use of a directed acyclic graph (Appendix A). To improve model convergence and interpretability, all covariates were centred to the covariate mean values and standardized to have a mean of zero and standard deviation of one. Selenium was included as a covariate standardised to 1 SD unit (26.66 μg/L) to estimate its association as an interaction term with mercury.

The following covariates and parameters were not considered to be confounding factors but were considered to improve model fit and precision: child sex, interaction terms between child age and other covariates, and non-linear terms for fish consumption variables. None of these were found to improve model fit as estimated by AIC and BIC [67].

Residual diagnostic plots indicated that linear modelling was a poor fit. To model non-linear change in growth across child age, splines with a linear, polynomial, and cubic function were evaluated. The spline parameterisation was found to be using linear splines on child age with knot points at 10 and 34 months.

The final covariate set of the mixed-effects model of child weight included mercury plus selenium, maternal age, parity, pre-pregnancy BMI, education level, smoking status, alcohol intake, white fish consumption, oily fish consumption, shellfish consumption, child age with linear spline knots at 10 and 34 months, a random intercept of participant and random slope of child age (further details in Appendix A). A second model was used for the purpose of extracting the interaction estimate between mercury and selenium. Results were extracted as normalised coefficients with 95% confidence intervals.

Linear multivariable models were used to estimate the association between mercury and static weight at each of the 10 time-points. To explore possible non-linearity, the 10 static weight models were repeated with subsets of the 10th Hg decile and 1st–9th Hg deciles. Each weight model was adjusted for the same covariates as the primary growth models, including child age to adjust for minor variation in the exact age of clinic attendance.

## 3. Results

### 3.1. Sample Characteristics

There were 593 children who (a) had at least one weight measurement between the age of 4 to 61 months from a Children in Focus clinic and (b) were born to mothers with a blood mercury measurement. After excluding very preterm births and multiple pregnancies, 544 children were included in the study (Figure 1).

The study subset was in general representative of the wider ALSPAC cohort (Table 1). There were slightly fewer participants smoking and more drinking alcohol during pregnancy in the subset, and they had higher levels of education.

Complete data for all selected covariates were available for 416 of the mother–child pairs, with the remaining 128 having at least one missing measurement. The most common missing variables were maternal pre-pregnancy BMI and alcohol consumed per week at 8 weeks’ gestation, both missing for 11% of mothers. There were no notable differences in the characteristics of those with or without missing data (full details in Appendix A). A directed acyclic graph illustrating how we understand these variables to be related to mercury and infant weight is available in Appendix A.

Pearson correlation coefficients between mercury and continuous covariates were calculated, and visualisations are available in Appendix A, including boxplots for categorical variables. Blood mercury and selenium levels were positively correlated (Pearson’s ρ = 0.20). Mean mercury increased with frequency of oily, white, and shellfish consumption and with the highest education qualification obtained, and was positively correlated with mother’s age (ρ = 0.24).

### 3.2. Exposure and Outcome Characteristics

In total, 544 children attended at least one Child in Focus clinic between the ages of 4 and 61 months, and the median number of time points attended was eight. The lowest attended time points were at 4 and 6 months with 381 children, and the highest was at 12 months with 484 children. The mean child weight at 4 months was 6.8 kg (SD: 0.8 kg) and at 61 months was 19.7 kg (SD: 2.8 kg). There were minimal differences between sexes: at 4 months, the median weights for boys and girls were 6.9 kg and 6.4 kg, respectively, and at 61 months they were 19.8 kg and 19.6 kg, respectively. Child growth trajectory and variance were similar for boys and girls (Figure 2), which was consistent with our model findings where the inclusion of child sex did not improve model performance at explaining variance in weight.

The median maternal blood mercury concentration was 1.9 µg/L, with an interquartile range of 1.1 µg/L (1st–3rd quartiles: 1.5 to 2.6 µg/L). The median time of blood collection was 11 weeks (IQR: 4 weeks), with 72% of samples collected in the first trimester (1–12 weeks) and 28% after. We did not find evidence of a difference in the distribution of mercury by time of collection (mean first trimester: 2.12 µg/L, mean after first trimester: 1.99 µg/L, two-sample *t*-test *p* = 0.99) and therefore included all samples in the subsequent analysis. The asymmetric distribution of mercury was left-skewed (Appendix A), and a similar distribution could be seen for selenium (median 109.7 µg/L).

### 3.3. Mercury, Selenium, and Change in Weight

There was little evidence of an association between mercury and postnatal growth after adjustment for potential confounders. A 1 SD unit (1.048 μg/L) change in blood mercury corresponded to 0.02 kg less weight gain per month, 95% CI: −0.10 to 0.06 kg. There was no evidence of an interaction effect between mercury and selenium on child weight gain (Table 2).

### 3.4. Mercury and Weight at Specific Time Points

When weight at each of the 10 clinic time points was modelled with mercury, there was no strong evidence of an association at any time point (Table 3). The 95% confidence intervals of the mercury coefficient overlapped with the null in each model, with the least overlap in the model of weight at 61 months, where 1 μg/L of mercury was estimated to change weight by 330 g (95% confidence interval: −3 g to 670 g).

There were indications of non-normally distributed residuals and heteroskedasticity in residual plots. To assess a possible non-linear relationship between mercury and weight, subsets of the 10th Hg decile (*n* = 53) and 1st–9th decile were modelled separately (*n* = 491) (Table 4). In the 10th decile subset, the estimate between mercury and weight was consistently negative across all ages but in all cases with a 95% confidence interval overlapping with zero. In the 1st–9th decile subset, the coefficient for mercury on weight was positive but overlapping the null. The point estimates between the 10th and 1st–9th decile subsets were consistently in opposite directions, which may indicate a threshold effect in those most exposed to mercury.

## 4. Discussion

This study of 544 mothers and children examined infant weight change between ages 4 and 61 months. Maternal mercury concentrations in whole blood during early pregnancy did not appear to be strongly associated with change in infant weight, and 95% confidence intervals overlapped with zero change. The coefficient estimate for 1 SD mercury of −0.02 kg per month would not amount to a considerable difference in weight at 61 months on an individual level but when applied to a whole population could be important. Mercury and selenium levels were positively correlated, but there was no indication of an interaction effect between the two metals on postnatal growth.

The results are consistent with those from a study of the Norwegian MoBa cohort [47]. Infant weight between 1 month and 8 years was modelled in a subset of 227 mother-child pairs in the 10th Hg decile (>2.23 µg/L). Maternal mercury (in µg/L) was estimated to change infant weight by −19 g per month, but with confidence intervals overlapping the null (95% CI: −102 g to 64 g) [47]. However, our study was smaller in sample size and not able to run a growth model in solely the 10th decile.

The second aim of this study was to model static infant weight and mercury at 10 intervals. When all children were included in a single linear multivariable model, we found no strong evidence that mercury was associated with a change in weight at any age. However, residual plots indicated that when modelling all children together the assumption of normality was not met. Model fit was improved by modelling the 1st–9th and 10th deciles separately, but results from these models continued to suggest no difference in weight change according to mercury exposure. There was weak evidence of a detrimental effect of mercury from the 10th decile mercury coefficients consistently being negative, a finding which replicates results from the previously mentioned Norway study, but in the Norwegian study 95% confidence intervals did not overlap with the null from age 18 months and onwards [47]. The MOCEH study in South Korea also reported a negative association between maternal mercury and weight at 24 months of age [46]. In both studies, the effect size and result certainty appeared to become stronger in older age groups. The size of the top Hg decile in our study was much smaller (*n* = 53) than in MoBa (*n* = 227) or MOCEH (*n* = 921), and this likely had a strong impact on the certainty of our results.

The concentrations of mercury found in maternal blood samples were intermediate to those in MoBa and MOCEH: the median in ALSPAC was 1.9 μg/L, which was higher than MoBa (1.03 μg/L) [47] but lower than MOCEH (early pregnancy: 3.5 μg/L) [46]. ALSPAC mercury concentrations were similar to those reported in other countries [68], including the USA, Canada, European countries, Iran, Turkey, and South Africa. However, there are countries where studies have reported higher concentrations, such as Greenland, Taiwan, Brazil, and Japan. There are few recommendations regarding the safe limit of blood mercury concentrations. In the context of acute mercury poisoning, a clinical review indicated that blood total Hg concentrations below 20 μg/L are considered acceptable [69]. However, a lower guidance value of 8 μg/L is recommended by Health Canada for women including those who are pregnant [70]. Guidance specific to pregnant women is limited, most likely because of the uncertain evidence regarding harms. A study located in the USA recommended that risks to foetal nervous development could be avoided with a reference dose level of 3.5 μg/L [71]. In this study most women were below the above thresholds (75% below 2.46 μg/L), but if the toxic expression of mercury is linear as some studies suggest [72], then any exposure should be minimized and could have an impact on growth at a population level.

The contribution of demographic characteristics and diet to mercury levels in ALSPAC have been explored extensively elsewhere [21]. In this subsample of the wider ALSPAC cohort, similar correlations were identified. From the selected covariates, maternal age, the highest level of education attained, and all forms of fish consumption were positively correlated with mercury concentrations. The correlation between fish consumption and mercury is possibly less than expected given the prominence placed in the literature on fish as a source of mercury exposure. A large proportion of the variance in mercury remained unexplained in the analysis by Golding et al. [21], which may reflect at least partly genetic variation in mercury metabolism. Despite the small geographic area (approximately 25 km^2^ [57]), there are local industries that could lead to variation in environmental exposure.

In summary, the results from this and prior studies [46,47] present weak evidence for an association between prenatal mercury exposure and postnatal growth. Although all studies report negative point estimates, in many cases confidence intervals overlap with the null and the range of possibilities for the true association remains wide. The uncertainty could be because postnatal growth is influenced by numerous other factors such as maternal and paternal smoking status [73], social class [73], child health [74], macronutrient intake [75], and exposure to other pollutants [76], to name a few, which increases model error terms as these factors are not fully measured. It may be that there is a non-linear effect of mercury that affects weight detectably at higher concentrations of mercury than those present in this study. Genetic variance in how mercury is absorbed, metabolised, and removed in either the mother and/or infant could mediate the effects of Hg exposure, which our analysis did not account for. The mother’s wider diet is also likely to play an important role in mediating mercury toxicity, and it could be that in this cohort there was adequate consumption of nutrients known to interact with mercury and mitigate its harms. This study investigated selenium and found no evidence of an interaction with mercury, but other key nutrients could include long-chain polyunsaturated fatty acids [77], vitamin D, iron, and zinc [78]. Consumption of these nutrients is likely linked to socio-economic status, which our models adjusted for, but they were not directly measured and together potentially could mitigate some of the risk from prenatal mercury exposure.

Our study had many strengths, including that child weight was measured at multiple time points, and we used mixed-effect models with linear splines split at knot points to account for the non-linear pattern of growth with child age. This allowed us to investigate the association between prenatal mercury exposure and change in weight, rather than only weight at specific time points. Additionally, the ALSPAC cohort database contains detailed measurements of family social and behavioural characteristics, which allowed us to adjust or control for previously identified confounders. Previous studies of mercury and birthweight have often failed to adjust for fish consumption [37], which is a potential positive confounder that could obscure any negative relationship because of the effects of beneficial nutrients in fish [79,80].

A limitation of this study is that we assume that maternal mercury concentrations can be used as a proxy for the infants’ prenatal mercury exposure. This study measured total mercury in whole blood, and the specific Hg species are not quantified. While there is strong evidence that certain forms of mercury can cross the placenta and accumulate in the developing child [81], this is not the case for inorganic mercury [82] and maternal variance of this compound may be irrelevant to prenatal exposure. Secondly, concentrations of micronutrients and other elements such as zinc [83], along with the infants’ mercury metabolism, are likely to affect the half-life of mercury and were also not measured. Studies that are able to include these elements as covariates will have greater model precision by accounting for more outcome variance, which would provide greater power to detect smaller effect sizes. Finally, some studies show that fish eating may moderate the effects of mercury toxicity [84], but the small size of this study made it impractical to stratify between fish eaters and non-fish eaters to investigate it.

## 5. Conclusions

In this study we examined maternal mercury concentration and repeated measures of weight during infancy using multilevel modelling. We found no strong evidence of an association, nor did selenium appear to interact with mercury levels to affect growth. An analysis that accounts for the full range of environmental contaminants and micronutrients that interact with mercury would be useful in confirming these findings.

## Figures and Tables

**Figure 1 toxics-11-00010-f001:**
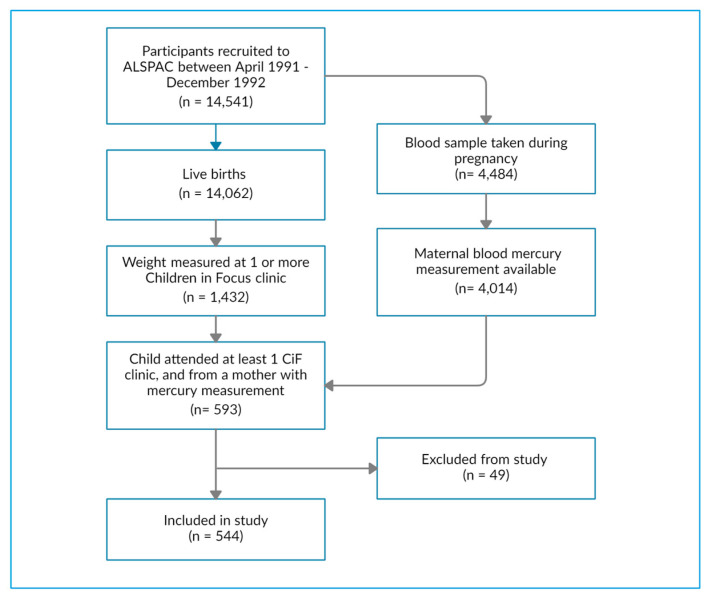
Flowchart of ALSPAC data available for this study.

**Figure 2 toxics-11-00010-f002:**
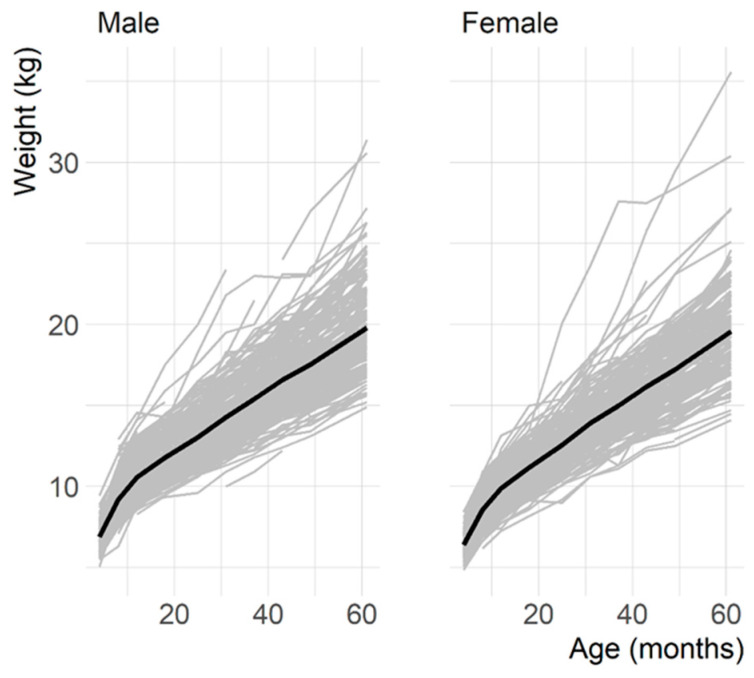
Weight trajectories from measurements of 544 children taken between 4 and 61 months of age, and mean weight highlighted in black.

**Table 1 toxics-11-00010-t001:** Characteristics of mother and child in ALSPAC and study samples. Median (IQR) or count (percent).

Variable	All ALSPAC	Mercury Samples	Mercury and Child Weight
**Mother**			
Number of births	14,062	3844	544
Maternal age (years)	28 (6)	28 (6)	28 (6.5)
**Parity**			
0	5585 (44.8%)	1597 (44.4%)	243 (46.0%)
1	4361 (35.0%)	1233 (34.3%)	171 (32.4%)
2+	2522 (20.2%)	765 (21.3%)	111 (21.0%)
**Highest level of education**			
O-level/below	7755 (64.6%)	2137 (61.6%)	306 (59.0%)
A-level/degree	4241 (35.4%)	1333 (38.4%)	212 (40.8%)
Pre-pregnancy BMI (kg/m^2^)	22.2 (3.9)	22.3 (4.0)	22.5 (4.1)
**Smoker at 8 weeks’ gestation**			
No	9348 (79.4%)	2803 (79.4%)	439 (85.9%)
Yes	2432 (20.6%)	727 (20.6%)	71 (13.9%)
**Drank alcohol at 8 weeks’ gestation**			
No	7932 (69.8%)	2351 (69.3%)	314 (64.5%)
Yes	3427 (30.2%)	1042 (30.7%)	172 (35.3%)
**Oily fish consumption**			
No	4961 (42.3%)	1463 (42.7%)	199 (38.9%)
Yes	6769 (57.7%)	1965 (57.3%)	312 (61.1%)
**White fish consumption**			
No	2157 (18.4%)	639 (18.6%)	89 (17.4%)
Yes	9573 (81.6%)	2789 (81.4%)	422 (82.6%)
**Shellfish consumption**			
No	9451 (80.6%)	2740 (79.9%)	391 (76.5%)
Yes	2279 (19.4%)	688 (20.1%)	121 (23.5%)
**Whole blood metal**			
Mercury (µg/L)	1.88 (1.17)	1.88 (1.17)	1.91 (1.13)
Selenium (µg/L)	108.4 (25.3)	108.4 (25.3)	109.7 (26.3)
**Child**			
Male	6934 (51.5%)	1985 (51.6%)	299 (55.0%)
Female	6535 (48.5%)	1859 (48.3%)	245 (45.0%)
Birthweight (g)	3440 (640)	3440 (650)	3500 (620)
Gestational age (days)	280 (14)	281 (14)	282 (11)

**Table 2 toxics-11-00010-t002:** Estimated growth (kg) per month between 4 and 61 months in the ALSPAC cohort using mixed effect modelling.

Predictor	Unadjusted (*n* = 544)		Adjusted (*n* = 544)	
	Coefficient	95% CI	Coefficient	95% CI
Mercury (1.048 μg/L)	−0.02	−0.09 to 0.06	−0.02	−0.10 to 0.06
Mercury * selenium interaction	0.00	−0.10 to 0.09	0.01	−0.09 to 0.12

Adjusted model: mothers’ age, parity, highest level of education, pre-pregnancy BMI, cigarettes per day, alcohol units per week, selenium, frequency of consumption of oily fish, white fish, and shellfish.

**Table 3 toxics-11-00010-t003:** Estimated associations between maternal blood mercury (μg/L) and child weight (kg), at 10 clinic visits.

Clinic Age	Mean Weight (kg)	*n*	Coefficient	95% CI
4 months	6.67	381	−0.01	−0.09 to 0.07
8 months	8.90	497	−0.02	−0.12 to 0.07
12 months	10.27	484	0.02	−0.09 to 0.14
18 months	11.49	445	0.03	−0.09 to 0.16
25 months	12.83	425	0.05	−0.10 to 0.20
31 months	14.11	438	0.10	−0.07 to 0.27
37 months	15.22	418	0.12	−0.07 to 0.30
43 months	16.41	413	0.14	−0.06 to 0.34
49 months	17.40	393	0.12	−0.11 to 0.35
61 months	19.69	381	0.33	−0.00 to 0.67

Adjusted for mothers’ age, child’s age at time of clinic visit, parity, highest level of education, pre-pregnancy BMI, cigarettes per day, alcohol units per week, selenium, frequency of consumption of oily fish, white fish, and shellfish.

**Table 4 toxics-11-00010-t004:** Estimated associations between deciles of maternal blood mercury (μg/L) and child weight (kg), at 10 clinic visits.

Clinic Age	Mean Weight (kg)	1st–9th Deciles Maternal Hg			10th Decile Maternal Hg		
		*n*	Coefficient	95% CI	*n*	Coefficient	95% CI
4 months	6.67	338	0.03	−0.11 to 0.17	43	−0.05	−0.33 to 0.22
8 months	8.90	444	0.05	−0.12 to 0.21	53	−0.18	−0.49 to 0.12
12 months	10.27	430	0.05	−0.13 to 0.23	54	−0.08	−0.45 to 0.30
18 months	11.49	397	0.06	−0.15 to 0.26	48	−0.22	−0.70 to 0.22
25 months	12.83	379	0.14	−0.10 to 0.39	46	−0.34	−0.86 to 0.17
31 months	14.11	388	0.12	−0.16 to 0.40	50	−0.20	−0.78 to 0.37
37 months	15.22	371	0.25	−0.06 to 0.60	47	−0.09	−0.42 to 0.60
43 months	16.41	362	0.22	−0.13 to 0.56	50	−0.21	−0.82 to 0.41
49 months	17.40	349	0.22	−0.15 to 0.60	44	−0.26	−1.03 to 0.51
61 months	19.69	337	0.50	−0.02 to 1.01	44	−0.01	−1.74 to 1.73

Adjusted for mothers’ age, child’s age at time of clinic visit, parity, highest level of education, pre-pregnancy BMI, cigarettes per day, alcohol units per week, selenium, frequency of consumption of oily fish, white fish, and shellfish.

## Data Availability

Access to ALSPAC data is through a system of managed open access (http://www.bristol.ac.uk/alspac/researchers/access/, accessed on 16 November 2022).

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
