# Peer review of "Prenatal Mercury Exposure and Infant Weight Trajectories in a UK Observational Birth Cohort"

_toxics, 2022, doi:10.3390/toxics11010010_

Round 1
Reviewer 1 Report
In this manuscript by Deck et al., the authors were interested in examining whether prenatal mercury exposure could lead to an increase in children’s weight, as claimed by many studies. For this, the authors made a retrospective analysis of the study conducted in 1990-1992. In the present study, the author claimed that there was no association between mercury exposure with children’s weight measured at different time intervals from 4 and 61 months. This observation was made based on human samples and pointed out all the limitations of the present study, which could be exciting aspects in looking forward.
Comments:
1. The authors are encouraged to include the permissible limit and the concentration of mercury found in different indices to highlight its environmental presence.
2. What was the rationale behind selecting prenatal mercury exposure from 9-18 weeks? Is there any study showing exposure during the time window is linked to children’s weight, irrespective of any environmental chemical exposure?
Author Response
We thank the reviewers for the positive feedback on our paper Prenatal mercury exposure and infant weight trajectories in a UK observational birth cohort, and the suggested revisions. We have adjusted the manuscript in response to the reviewers feedback, and changes are listed below.
- The authors are encouraged to include the permissible limit and the concentration of mercury found in different indices to highlight its environmental presence.
There are few recommendations which are (a) directed at pregnant women, and (b) report blood mercury thresholds. The following has been added to the discussion:
“There are few recommendations regarding the safe limit of blood mercury concentrations. In the context of acute mercury poisoning, a clinical review indicated that blood total Hg concentrations below 20 μg/l are considered acceptable [69]. However, a lower guidance value of 8 μg/l is recommended by Health Canada for women including those who are pregnant [70]. Guidance specific to pregnant women is limited, most likely because of the uncertain evidence regarding harms. A study located in the USA recommended that risks to fetal nervous development could be avoided with a reference dose level of 3.5 µg/l [71]. In this study most women were below the above thresholds (75% below 2.46 µg/l), but if the toxic expression of mercury is linear as some studies suggest [72], then any exposure should be minimized and could have an impact on growth at a population-level.”
- What was the rationale behind selecting prenatal mercury exposure from 9-18 weeks? Is there any study showing exposure during the time window is linked to children’s weight, irrespective of any environmental chemical exposure?
We were interested in the effect of prenatal exposure to mercury, and in this pre-existing cohort, this was the timing that samples were available from. We investigated whether there were differences in mercury levels based on timing of blood collection, and have added the following:
“The median time of blood collection was 11 weeks (IQR: 4 weeks), with 72% of samples collected in the first trimester (1-12 weeks) and 28% after. We did not find evidence of a statistical difference in the distribution of mercury by time of collection (mean first trimester: 2.12 µg/l, mean after first trimester: 1.99 µg/l, two-sample t-test p = 0.99), and therefore included all samples in the subsequent analysis.”
Additionally, the following paragraph was added to the introduction to clarify the background and rationale to this study:
“It is possible that prenatal mercury exposure may also impact growth in the postnatal pe-riod, due to the slow clearance rate of mercury. The biological half-life of mercury – the speed at which concentrations are halved – is estimated to be between 30 and 120 days for methylmercury [43] and 30 to 60 days for inorganic mercury [44]. However, in children metabolic rates may differ, and in certain tissue such as the brain retention is believed to be much longer [45]. The postnatal environment may differ from the prenatal in ways which are less protective of the toxic effects of mercury, and the impact on growth may be more detectable. For example, it is possible after birth for the intake of long-chain polyun-saturated fatty acids to fall [45], a nutrient protective against Hg toxicity [45].”
Reviewer 2 Report
Dear authors,
thank for this very interesting article „Prenatal mercury exposure and infant weight trajectories in a UK observational birth cohort”.
The article is of general interest. Mercury is toxic substance of great concern, and more information is needed, since the evidence of the role of mercury for health in growing. So in this case, evidence that prenatal exposure to mercury does not have a clinically significant influence on growth development is very interesting. The paper is very well written. And the ALSPAC study is really such a great source!
Minor comments:
Table 3 and 4 have the same heading. Needs to be corrected.
Figure 1 in the supplement is really good, may be you can move it to the main part.
Good luck with the revisions.
Author Response
We thank the reviewers for the positive feedback on our paper Prenatal mercury exposure and infant weight trajectories in a UK observational birth cohort, and the suggested revisions. We have adjusted the manuscript in response to the reviewers feedback, and changes are listed below.
- Table 3 and 4 have the same heading. Needs to be corrected.
The Table 4 heading has been corrected: “Table 4. Estimated associations between deciles of maternal blood mercury (μg/l) and child weight (kg), at 10 clinic visits.”
- Figure 1 in the supplement is really good, may be you can move it to the main part.
We agree that the directed acyclic graph is a useful tool to show the relationship between variables in this study. However, the diagram is quite complex because of the large number of variables and interlinked relationships. To include it in the main paper would most likely require a more detailed explanation and may overall distract readers from the main findings of the paper. As such we have chosen to keep it accessible in the Supplementary Materials, adding the following line to the results:
“A directed acyclic graph illustrating how we understand these variables to be related to mercury and infant weight is available in Supplementary Figure S1.”
Reviewer 3 Report
This manuscript describes analysis of data collected in the ALSPAC study that looks at maternal blood mercury levels and subsequent child weight trajectory up to 61 months of age. The authors report that no significant associations between blood mercury levels and infant weight changes. The authors ascribe these results to the Hg levels in this cohort having minimal biological impact and the effect being too small to be clinically relevant or detectable.
The authors describe two studies (the MOCEH and MoBa studies) that have assessed prenatal mercury and infant growth, one of which – the MoBa was larger and had >2000 mother-child pairs. The result of the MoBa study were reported as indicating that there was a reduction in child’s weight growth trajectory at 18 months in the top decile of prenatal mercury exposure. The maternal Hg levels in this study seem comparable to the MoBa study but the number of mother-child pairs is much smaller (being ~25% of the MoBa study). One potential reason then for the lack of any association in this ALSPAC study is that the study was insufficiently powered to detect the effect seen in the MoBa study. What power calculations did the authors carry out and indeed what was the power of their study?
A second reason for the lack of effect may have been exposure misclassification. The directed acyclic graph (supplementary Figure S1) indicates that mercury levels in the first trimester are important. The authors indicate that 93% of samples were collected before 18 weeks (line 109). How many were collected in the first trimester? As there is evidence that maternal mercury blood levels vary with trimester and, in particular, may be lower in the second and third trimesters, there may been exposure misclassification due to the inclusion of maternal Hg levels measured in the second trimester. Have the authors restricted their analysis to include only those samples collected in the first trimester?
Can the authors also clarify the following points:
1. The abstract indicates that “Dietary changes after birth may make it easier to detect a negative impact on infant growth”. It is unclear why this sentence is here as diet after birth is not investigated.
2. Lines 198-200 are in the template and should have been deleted
3. The reasons behind the exclusion of 49 children from the study (Figure 1). Is it due to very preterm births and multiple pregnancies?
4. What does the dark line in Figure 2 signify?
5. The meaning of different coloured symbols and lines in Figure S1
Author Response
We thank the reviewers for the positive feedback on our paper Prenatal mercury exposure and infant weight trajectories in a UK observational birth cohort, and the suggested revisions. We have adjusted the manuscript in response to the reviewers feedback, and changes are listed below.
- What power calculations did the authors carry out and indeed what was the power of their study?
We were unable to conduct a power calculation prior to the study, because at the time there were no comparable studies published (the MoBA study was published after our analysis was complete). Without comparable studies, the potential effect size and variance were unknown, and without these parameters a power calculation would not be informative.
We have not included a post-hoc power calculation because we can already ascertain the result of this from our results – a non-significant result will inevitability be determined to be underpowered in a post-hoc power calculation. We agree with the reviewers suggestion that power may be one explanation for the lack of a detected association, and touch on this in the discussion section [373-375].
“Secondly, concentrations of micronutrients and other elements such as zinc [83], along with the infants’ mercury metabolism, are likely to affect the half-life of mercury and were also not measured. Studies that are able to include these elements as covariates will have greater model precision by accounting for more outcome variance, which would provide greater power to detect smaller effect sizes. Finally, some studies show that fish eating may moderate the effects of mercury toxicity [84], but the small size of this study made it impractical to stratify between fish eaters and non-fish eaters to investigate it.”
- A second reason for the lack of effect may have been exposure misclassification. The directed acyclic graph (supplementary Figure S1) indicates that mercury levels in the first trimester are important. The authors indicate that 93% of samples were collected before 18 weeks (line 109). How many were collected in the first trimester? As there is evidence that maternal mercury blood levels vary with trimester and, in particular, may be lower in the second and third trimesters, there may been exposure misclassification due to the inclusion of maternal Hg levels measured in the second trimester. Have the authors restricted their analysis to include only those samples collected in the first trimester?
We have explored the distribution of Hg by age at time of collection, but did not find strong of evidence a difference in mercury levels by time of collection in this cohort. Therefore we did not consider it necessary to restrict to first trimester only. The following lines have been added to the results:
“The median time of blood collection was 11 weeks (IQR: 4 weeks), with 72% of samples collected in the first trimester (1-12 weeks) and 28% after. We did not find evidence of a difference in the distribution of mercury by time of collection (mean first trimester: 2.12 µg/l, mean after first trimester: 1.99 µg/l, two-sample t-test p = 0.99), and therefore included all samples in the subsequent analysis.”
- The abstract indicates that “Dietary changes after birth may make it easier to detect a negative impact on infant growth”. It is unclear why this sentence is here as diet after birth is not investigated.
This sentence was replaced with a more general statement:
“Differences in nutrition, exposures, and the living environment after birth may make it easier to detect a negative impact from mercury toxicity on infant growth.”
- Lines 198-200 are in the template and should have been deleted
These lines have been deleted.
- The reasons behind the exclusion of 49 children from the study (Figure 1). Is it due to very preterm births and multiple pregnancies?
These were children excluded for being either (a) born very prematurely (<33 weeks) or (b) multiple pregnancies as stated on lines 205/206. We only report the total excluded, and not the breakdown for (a) and (b), because we are not permitted to publish very small numbers to protect the anonymity of participants.
- What does the dark line in Figure 2 signify?
The heading of figure 2 has been modified to clarify.
“Weight trajectories from measurements of 544 children taken between 4 and 61 months of age, and mean weight highlighted in black.”
- The meaning of different coloured symbols and lines in Figure S1
A note has been added to explain the diagram:
“Green circles represent the exposures of interest. Blue circles with “I” represent the outcome of interest. Red circles are ancestors of both the exposure and outcome, and therefore considered confounding variables. Gray circles are latent variables. The green arrows represent the causal pathway from exposure to outcome, and the red arrows represent confounding pathways.”